# High-Throughput Production of Heterogeneous RuO₂/Graphene Catalyst in a Hydrodynamic Reactor for Selective Alcohol Oxidation

**Jae-Min Jeong** [1,†] , **Se Bin Jin** [2,†] , **Jo Hee Yoon** [2] , **Jae Goo Yeo** [2] , **Geun Young Lee** [2] ,
**Mobina Irshad** [2] , **Seongwoo Lee** [2] , **Donghyuk Seo** [1] , **Byeong Eun Kwak** [1] , **Bong Gill Choi** [2,*] ,
**Do Hyun Kim** [1,*] and **Jung Won Kim** [2,*]

[1]  Department of Chemical & Biomolecular Engineering, Korea Advanced Institute of Science and Technology,
   Daejeon 34141, Korea; jmjeong@kaist.ac.kr (J.-M.J.); hyuk1553@kaist.ac.kr (D.S.);
   byeongeunkwak@kaist.ac.kr (B.E.K.)
[2]  Department of Chemical Engineering, Kangwon National University, Samcheok 25913, Korea;
   tpqls1230@kangwon.ac.kr (S.B.J.); joheeyoon@kangwon.ac.kr (J.H.Y.); jaegoo@kangwon.ac.kr (J.G.Y.);
   aedd123@kangwon.ac.kr (G.Y.L.); mobinachemist@kangwon.ac.kr (M.I.); seongwoo8135@kangwon.ac (S.L.)
*  Correspondence: bgchoi@kangwon.ac.kr (B.G.C.); dokim@kaist.ac.kr (D.H.K.);
   jwemye@kangwon.ac.kr (J.W.K.); Tel.: +82-33-570-6543 (J.W.K.)
†  These authors were equally contributed this paper.

**Abstract:** We report on the high-throughput production of heterogeneous catalysts of RuO₂-deposited graphene using a hydrodynamic process for selective alcohol oxidation. The fluid mechanics of a hydrodynamic process based on a Taylor–Couette flow provide a high shear stress field and fast mixing process. The unique fluidic behavior efficiently exfoliates graphite into defect-free graphene sheets dispersed in water solution, in which ionic liquid is used as the stabilizing reagent to prevent the restacking of the graphene sheets. The deposition of RuO₂ on a graphene surface is performed using a hydrodynamic process, resulting in the uniform coating of RuO₂ nanoparticles. The as synthesized RuO₂/IL–graphene catalyst has been applied efficiently for the oxidation of a wide variety of alcohol substrates, including biomass-derived 5-hydroxymethylfurfural (HMF) under environmentally benign conditions. The catalyst is mechanically stable and recyclable, confirming its heterogeneous nature.

**Keywords:** heterogeneous catalyst; hydrodynamic process; selective alcohol oxidation; biomass-derived HMF; graphene; ruthenium oxide

## 1. Introduction

With increasing concerns of environmental pollution and the exhaustion of fossil fuel resources, renewable materials (biomass) are of great interest in various applications ranging from biofuels to important commodity chemicals [1,2]. In particular, 2-furaldehyde (Furfural), 5-hydroxymethylfurfural (HMF), and their derivative chemicals have gained a great deal of attention as important platform compounds in the biorefinery process because of their ability to produce renewable and non-petroleum chemical feedstocks of solvents, polymers, and fuels [3–6]. Furfural and HMF have been generally derived from homogeneous catalytic reactions such as hydrolysis, dehydration, and selective alcohol oxidation, which exist in large quantities in xylan and sucrose [3,6]. The utilization of homogeneous catalysts and oxidizing reagents provide a high yield and fast reaction rates for the conversion and production of chemical platforms. However, it is difficult to separate homogeneous catalysts from reaction media, resulting in an increase in production cost [7]. In addition, homogeneous

catalyst reactions using these oxidants produce inevitable heavy metal by-products, which may cause significant environmental pollution [8]. Heterogeneous catalysts have been also developed for alcohol oxidation reactions because of removing or substantially reducing pollution and undesirable by-products from both the chemical and refining processes [9–11]. To date, various types of heterogeneous catalysts have been extensively investigated, including Au, Ag, Pt, Pd, and Ru deposited on chemically stable supports (e.g., silica and zeolite), which show improvements in the catalytic activity [12]. Among them, carbon supports-based novel metal catalysts (e.g., palladium, copper, and ruthenium) are better with respect to green and cost-effective chemistry principles, such as atom economy, environmentally oxidizing agents, and simplified reaction processes [13]. The catalysts' activity relies on mainly the surface characteristics and surface area of carbon substrates, which enable the easy access of reactant species.

Graphene is a single layer of carbon atoms arranged in a two-dimensional lattice, and has been attractive as a support of heterogeneous catalysts because of its large theoretical specific surface area of 2630 $m^2$/g, high mechanical strength, excellent chemical and thermal stability, and the high adsorption ability of its target molecules [14]. In particular, the preparation of graphene oxide (GO), which is derived from the chemical oxidation and exfoliation of graphite, expanded its application for catalytic supports because of its relatively higher production rate compared to pristine graphene and surface-oxygenated functional groups for the binding sites of catalytic active materials. For instance, Tsubaki et al. reported that a Pt nanoparticle-deposited GO catalyst enhanced the yield of hydrogenolysis reaction of cellulose site compared to $Pt/SiO_2$ because of the stable dispersion and large surface area of Pt/GO [15]. Chaudhari et al. also reported on the fast and selective hydrogenolysis reaction of CuPd/GO as a catalyst [16]. However, the preparation of GO requires multiple complicated and time-consuming procedures with toxic oxidizing agents. As a desired carbon support for heterogeneous catalysts, graphene materials should (1) have the intrinsic structure of a single carbon layer with a high surface area and (2) be produced in large quantities using a sustainable and eco-friendly method.

Recently, our group reported an eco-friendly hydrodynamic process for the preparation of defect-free graphene sheets dispersed in water using the strong shear force and high mixing power of Taylor–Couette (T.C.) flow [17,18]. In this work, we synthesized a heterogeneous catalyst of $RuO_2$/defect-free graphene using a hydrodynamic synthesis based on T.C. flow. During the synthesis of the $RuO_2$/graphene catalyst, ionic liquid (IL) was added as an exfoliating and stabilizing reagent for graphene sheets. ILs provided abundant biding sites of $RuO_2$ and enabled the uniform and conformal deposition of $RuO_2$ onto a graphene surface. The resultant $RuO_2$/IL–graphene was used as a heterogeneous catalyst for the selective oxidation of biomass-derived HMF to 2,5-diformylfuran (DFF) and furfuryl alcohol to furfural. The scope of this reaction is used for the oxidation of other primary alcohols, secondary alcohols, substituted alcohols, and heteroatomic alcohols in wide applications. The catalyst showed excellent conversion and selectivity for various alcohols in molecular oxygen as a sole oxidant. A recycle test confirmed the retention of the catalytic activity of the prepared catalyst over five cycles of benzylalcohol and HMF.

## 2. Results and Discussion

The preparation of IL–graphene and $RuO_2$/IL–graphene using a hydrodynamic method in a T.C. flow reactor is presented schematically in Figure 1a. The T.C. flow is generated in a small gap of two concentric inner and stationary cylinders at a critical rotational speed of 2000 rpm. The T.C. flow has uniform Taylor vortexes with efficient and fast mixing flow (Figure 1b,c) [19]. These unique T.C. flow characteristics lead to a strong wall shear force. The fluid mechanics of T.C. flow, such as shear, pressure, and collision stressed graphite, and thus exfoliated it into mono or few-layer graphene sheets with a high-level graphitic structure. During this exfoliation process, IL additives attached to exfoliated graphene sheets through the π-π interactions between them. The IL functionalities on graphene surface prevent the reaggregation of graphene because of the high hydrophilic nature of

ILs. In this work, we selected 1-ethyl-3-methylimidazolium ([EMIM][BF$_4$]) as IL because [EMIM][BF$_4$] enables a higher exfoliation yield of graphene compared to other ILs. The optimization of ILs was investigated in our previous reports [17]. Graphite (10 mg/mL) dispersed in a mixture of water/IL (0.15 vol %) was fed into a T.C. reactor to prepare IL–graphene sheets. After exfoliating for 30 mins, the synthesis of RuO$_2$/IL–graphene was carried out by injecting Ru precursors (0.1 M) into the T.C. reactor. The exfoliation yield and final concentration of IL–graphene were 48.9% and 13.4 mg/mL, which were obtained from the centrifugation (420 *g*) of exfoliated IL–graphene dispersions. Noticeably, the deposition time of RuO$_2$ onto the IL–graphene surface was less than five minutes, which is lower than other conventional synthetic methods, including hydrothermal, thermal stirring, and ball milling. After hydrodynamic synthesis, the final products of RuO$_2$/IL–graphene powder were obtained by washing and filtering the RuO$_2$/IL–graphene dispersions. The percent amount of Ru on RuO$_2$/IL–graphene was 1.5 wt %. The production rate of RuO$_2$/IL–graphene was 1.3 g/h. The hydrodynamic synthesis is a straightforward and fast process to effectively exfoliate a large quantity of graphite and prepare a high-quality heterogeneous catalyst of RuO$_2$/IL–graphene.

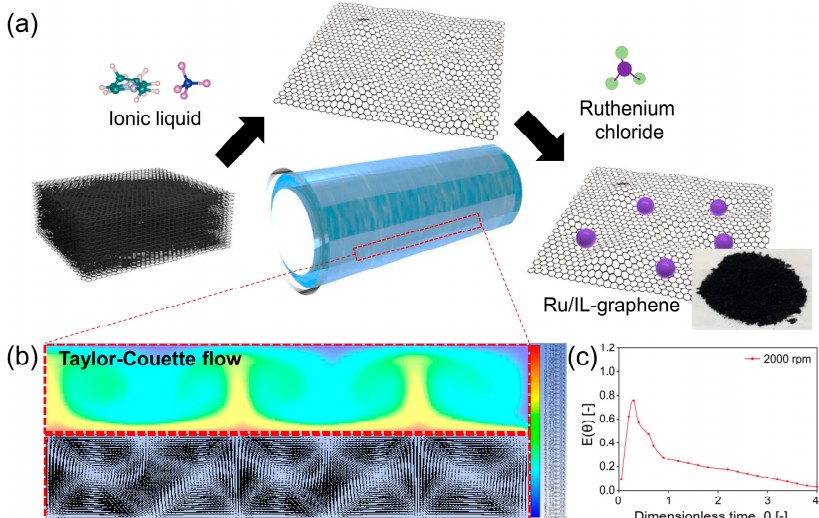

**Figure 1.** (**a**) Schematic illustration of the hydrodynamic synthesis for ionic liquid (IL)–graphene and RuO$_2$/IL–graphene. Photograph image of RuO$_2$/IL–graphene powders. (**b**) Computational fluid dynamics (CFD) data of *xz* axis and tangential velocity diagram and contour at 2000 rpm. (**c**) Residence time distribution (RTD) profiles measured at 2000 rpm.

The obtained IL–graphene and RuO$_2$/IL–graphene powder samples can be easily redispersed in water solution because of the presence of hydrophilic IL functionalities, showing excellent stable dispersions over one month (inset of Figure 2a,b). A hydrodynamic process using fluid mechanics enabled graphite to efficiently exfoliate and created large-sized graphene sheets. The average lateral size of IL–graphene was evaluated to 50-100 μm by counting 100 sheets of IL–graphene using scanning electron microscope (SEM) images (Figure 2a). This high lateral size value is much higher than those obtained from other exfoliating methods, including sonication, homogenizer, and the ball milling method [20]. Obviously, the transmission electron microscope (TEM) image showed the folded two-dimensional (2D) sheets, indicating a few-layer graphene. A high-quality graphitic structure was demonstrated by observation of high-resolution TEM and selected area electron diffraction (SAED) pattern measurement (Figure 1b). Clear carbon lattices with a honeycomb structure were observed. The corresponding hexagonal carbon lattice patterns of the SAED are the exhibition of sp$^2$-bonded carbon frameworks [21]. The hydrodynamic method provided a uniform deposition of RuO$_2$ nanoparticles onto the IL–graphene surface (Figure 2b). The elemental mapping measurement of ruthenium and carbon elements show overlapped images of the two elements, indicating the uniform distribution of RuO$_2$ nanoparticles. As shown in a histogram of

nanoparticle size, the average size of $RuO_2$ nanoparticles was $1.4 \pm 1.1$ nm. A lattice fringe of $RuO_2$ nanoparticles was observed in a high-resolution TEM image, indicating a crystalline structure of $RuO_2$ nanoparticles. Crystalline structures of IL–graphene and $RuO_2$/IL–graphene were investigated using Raman spectroscopy and X-ray Diffraction (XRD) measurement. Figure 3a shows a Raman spectrum of IL–graphene with D and G bands. The ratio value (0.45) of $I_D$ and $I_G$ was lower than GO [22], indicating that IL–graphene has a high-level graphitic structure. An XRD pattern of $RuO_2$/IL–graphene shows the formation of $RuO_2$ on IL–graphene. Compared to graphite, a significantly decreased d-spacing peak at $24°$ for IL–graphene indicates an excellent exfoliation state for the graphene sheets [23].

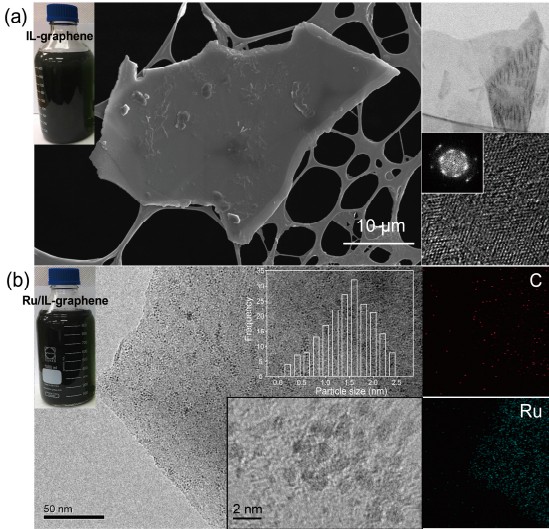

**Figure 2.** (**a**) SEM, TEM, and HR-TEM images of IL–graphene (the left inset is a photographic image of IL–graphene dispersion, and the right inset is selected area electron diffraction (SEAD) pattern of IL–graphene). (**b**) TEM and elemental mapping (C and Ru) images (left inset is photograph image of $RuO_2$/IL–graphene dispersion and the right inset is HR-TEM image of $RuO_2$/IL–graphene).

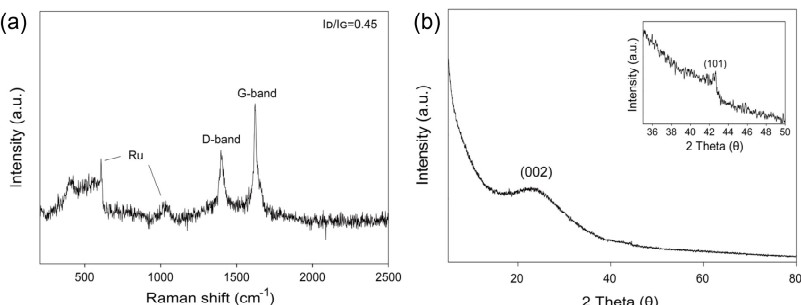

**Figure 3.** (**a**) Raman spectrum and (**b**) XRD pattern of $RuO_2$/IL–graphene (Inset is high-resolution of the XRD pattern).

The chemical state of $RuO_2$/IL–graphene was further investigated using XPS spectroscopy. Figure 4a shows a survey scan of $RuO_2$/IL–graphene. The X-ray photoelectron spectroscopy (XPS) spectrum shows carbon signals for graphene and ruthenium and oxygen signals for ruthenium oxide, verifying the co-existence of $RuO_2$ and graphene. The high-resolution spectra of Ru 3d of $RuO_2$/IL–graphene also overlaps with the C1s peak (at 284.5 eV). The Ru $3d_{5/2}$ spectrum corresponds to the binding energy of $Ru^{4+}$ [24], which is indicative of the presence of the $RuO_2$ phase in $RuO_2$/IL–graphene. A number of transition metals have been used for a long time to prepare the supported catalysts for oxidation reactions [25,26]. Among these metals, ruthenium gained a huge interest because of its unique characteristics [27,28]. In the given protocol, a number of ruthenium catalysts were tested for the conversion of benzylalcohol to benzaldehyde. Along with our catalyst, $RuO_2$/IL–graphene, several other $RuO_2$-supported catalysts were subjected to oxidation reaction,

but in most of the cases, the conversion rate was not comparable with the catalyst, although the selectivities were good for other catalysts (Table 1). So, the $RuO_2$/IL–graphene was applied for biomass-derived HMF and furfuryl alcohol along with another wide range of alcohols. HMF is more likely to oxidate into DFF under ruthenium-supported catalysts in moderate conditions with only molecular oxygen [29].

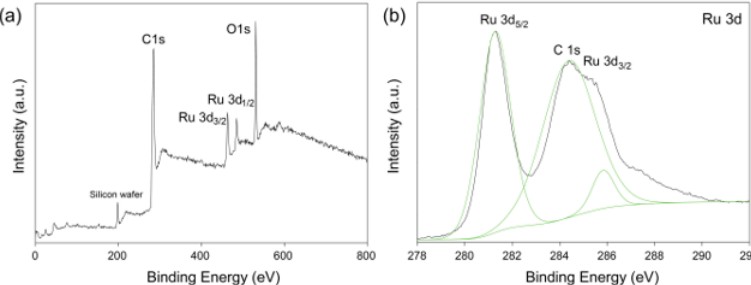

**Figure 4.** XPS spectra of $RuO_2$/IL–graphene: (**a**) Survey scan and (**b**) core level of Ru 3d and C 1s.

**Table 1.** Selective aerobic alcohol oxidation in the presence of various ruthenium catalysts.

| Entry | Catalyst | Conv. [b] (%) | Select. (%) |
|---|---|---|---|
| 1 | $RuO_2$/IL-graphene [a] | >99 | >99 |
| 2 | IL-graphene [b] | No reaction | No reaction |
| 3 | $RuCl_3 \cdot nH_2O$ [c] | 5 | <5 |
| 4 | $Ru(OH)_x$/Bulk C [d] | 44 | >99 |
| 5 | $Ru(OH)_x$/SNP | 4 | >99 |
| 6 | $Ru(OH)_x$/$MoS_2$ | >99 | >99 |
| 7 | $Ru(OH)_x$/$Al_2O_3$ | 52 | >99 |
| 8 | $Ru(OH)_x$/$MnO_2$ | 82 | >99 |
| 9 | $Ru(OH)_x$/$CeO_2$ | 45 | >99 |
| 10 | $Ru(OH)_x$/$ZrO_2$ | 23 | >99 |
| 11 | No catalyst | No reaction | No reaction |

[a] Reaction conditions: $RuO_2$/IL-graphene (Ru = 1.5 wt%), benzylalcohol (0.5 mmol), toluene (3 mL), 100 °C, $O_2$ flow (1 atm), 2 h. Conversion and selectivity were calculated by GC and GC-MS analysis by using biphenyl as an internal standard. [b] Graphene support for ruthenium. [c] Ru precursor. [d] Ru supported on bulk carbon. [e] Ru supported on silica nanoparticles.

Table 2 shows the conversion of various alcohol substrates under our optimized conditions with the $RuO_2$/IL–graphene catalyst. The biomass-derived HMF was oxidized under given conditions, and 60% conversion was obtained with >99% high selectivity (entry one). Furfural was obtained from fufuryl alcohol with a 70% yield (entry two). Benzylalcohol demonstrated >99% yield and conversion with more than 99% selectivity (entry three). Various substituted alcohols showed different reaction rates depending upon the functional groups attached to them. Substituted alcohols with methyl and methoxy electron-donating groups finished the reaction just in one hour and 10 min, respectively (entries four and five). Chloro- and nitro-substituted alcohols gave the desirable results after a long time due to the electron-withdrawing effect of the respective groups (entries six and seven). In the reaction of heteroatomic alcohol, sulfur containing one provided an excellent yield of the corresponding aldehyde (entry eight). The present system could oxidize secondary cyclic aliphatic alcohol in high yield (entry nine). A recycle test was performed to check the mechanical stability of the $RuO_2$/IL–graphene catalyst for the same oxidation reaction of benzylalcohol and HMF. In particular, the catalyst showed almost 60% catalytic activity for the oxidation of the HMF biomass intermediate over five cycles under the same reaction conditions. The recycling ability test is shown in Figure 5.

**Table 2.** RuO$_2$/IL–graphene catalyzed selective oxidation of several alcohols with O$_2$ as an oxidant.

| Entry | Substrate | Time (h) | Conv. (%) | Yield (%) | Select. (%) | Product |
|---|---|---|---|---|---|---|
| 1 | | 12 | 60 | 60 | >99 | |
| 2 | | 12 | 70 | 70 | >99 | |
| 3 | | 2 | >99 | >99 | >99 | |
| 4 | | 1 | >99 | >99 | >99 | |
| 5 | | 0.16 (10 min) | >99 | >99 | >99 | |
| 6 | | 18 | >95 | >95 | >99 | |
| 7 | | 6 | >93 | >93 | >99 | |
| 8 | | 8 | >99 | >99 | >99 | |
| 9 | | 6 | >92 | >92 | >99 | |

Reaction conditions: RuO$_2$/IL-graphene (Ru = 1.5 wt%, 10 mg), substrate (0.5 mmol), toluene (3 ml), 100 °C, O$_2$ flow (1 atm), Conversion and selectivity were determined by GC and GC-MS analysis and 1,4-biphenyl as internal standard.

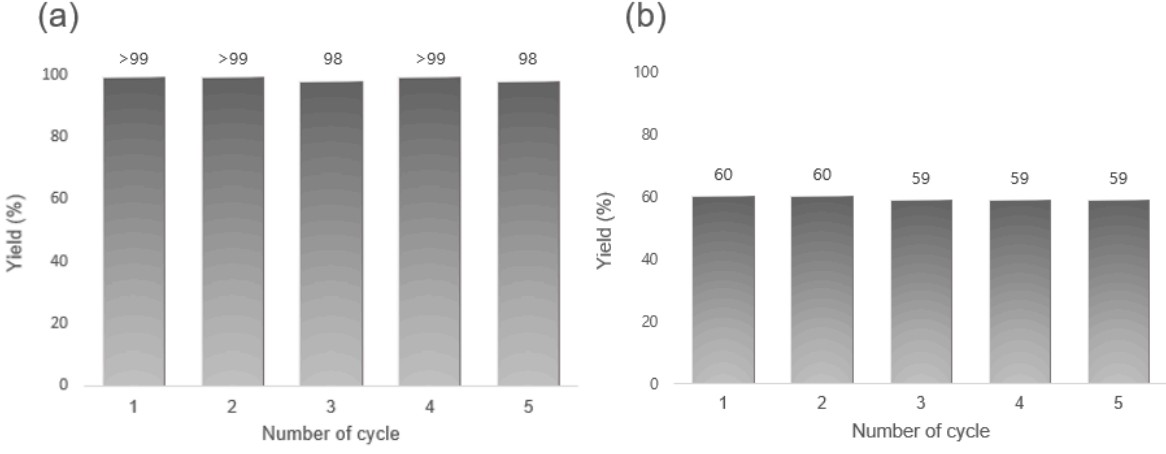

**Figure 5.** Recycle ability test of selective oxidation for (**a**) benzylalcohol and (**b**) 5-hydroxymethylfurfural (HMF).

## 3. Materials and Methods

### 3.1. Chemicals

Distilled water (DI water) was prepared by Milli-Q water (18.2 MΩ/cm resistivity). Graphite powder (<200 μm), ruthenium (III) chloride hydrate, sodium sulfate, 5-hydroxmethylfurfural, and furfuryl alcohol were obtained from Sigma-Aldrich (St. Louis, MO, USA). [EMIM][BF$_4$] were purchased from C-TRI company (Namyangju-si, Korea). All of the other reagents and solvents were used as received without further purification.

### 3.2. Preparation of IL–Graphene and RuO$_2$/IL–Graphene

IL–graphene was prepared using a hydrodynamic reactor. The reactor comprised concentric inner (rotational) and (stationary) outer cylinders. The hydrodynamic reactor was made of stainless steel with a cylinder gap distance of 0.5 mm. First, graphite was soaked into DI water (10 mg/mL) and 0.15 vol % of IL was additionally mixed. The graphite and IL mixture was injected into a hydrodynamic reactor, and then graphite was exfoliated in the reactor at 2000 rpm for 30 min. After exfoliation, supernatants and residual IL were removed by centrifugation at 5000 *g* for 60 min. A few layers of IL–graphene dispersion were obtained by removing unexfoliated graphite by centrifugation at 420 *g* for 150 min. To prepared RuO$_2$/IL–graphene catalysts, 0.01 M of ruthenium chloride hydrate (0.5 mL) was injected into the reactor after exfoliating graphite, and then 0.1 M of sodium sulfate (0.5 mL) was subsequently added to the reactor at 2000 rpm. After five minutes, RuO$_2$/IL–graphene was washed and filtered with ethanol and DI water.

### 3.3. Material Characterization

SEM image was investigated by a field emission scanning electron microscope (S-4800). High- and low-resolution TEM images were obtained using a field emission transmission electron microscope (JEM2100F, JEOL Ltd., Tokyo, Japan) operated at 300 kV. Raman microscopy (ARAMIS, HORIBA Jobin Yvon, HORIBA, Anyang-Si, Korea) was performed on a 514-nm laser of excitation. XRD data were obtained on RigakuD/MAX-2500 (40 kW, Tokyo, Japan) with a $\theta/\theta$ goniometer. XPS data was performed by Thermo MultiLab 20000 (Thermo Fisher Scientific, Daejeon, Korea).

### 3.4. Catalytic Reaction

For the HMF, furfuryl alcohol, and other alcohol substrates, 0.5 mmol of alcohols were added separately in a glass reactor containing a magnetic bar followed by the addition of three mL of toluene as a solvent. After that, the RuO$_2$/IL–graphene catalyst (Ru: 1.5 wt %, 10 mg) was added, and the temperature was raised up to 100 °C under one atm pressure of O$_2$. The mixture was vigorously stirred, and samples were taken at regular intervals. The yield and product selectivity were analyzed by gas chromatography (GC), and all of the products such as DFF, furfural, and aldehydes were identified by comparison of their [1]H, [13]C NMR, and mass spectra with original samples or literature data. After the first reaction was complete, the spent RuO$_2$/IL–graphene catalyst could be easily separated from the reaction mixture by filtration. The isolated catalyst was washed with water simply and then dried in vacuo before being recycled. These recycling procedures were repeated several times under same reaction conditions.

## 4. Conclusions

We synthesized a heterogeneous RuO$_2$/IL–graphene catalyst using a hydrodynamic method for highly active and selective alcohol oxidation. The developed hydrodynamic process provided highly exfoliated and functionalized IL–graphene sheets dispersed in water and enabled the fast and uniform deposition of RuO$_2$ nanoparticles on graphene supports. The defect-free carbon structure of graphene enhances the physicochemical and thermal stability of the RuO$_2$ catalyst, while IL provides abundant biding sites for the deposition of RuO$_2$ without the aggregation of RuO$_2$ nanoparticles. Furthermore, the RuO$_2$/IL–graphene catalyst was sufficiently effective to carry out the selective oxidation of biomass-derived HMF and furfuryl alcohol to corresponding aldehydes along with a wide range of other alcohol substrates, and is mechanically stable to be recycled over several catalytic cycles.

**Author Contributions:** Conceptualization, J.W.K. and B.G.C.; Data curation, J.H.Y.; Formal analysis, S.B.J., M.I. and D.H.K.; Funding acquisition, J.W.K., D.H.K. and B.G.C.; Investigation, G.Y.L., D.S. and B.E.K.; Methodology, J.-M.J.; Project administration, J.W.K.; Supervision, D.H.K.; Validation, J.Y.; Visualization, S.L.; Writing—original draft, J.-M.J. and S.B.J; Writing—review & editing, J.W.K. and B.G.C.

**Funding:** This work was supported by the National Research Foundation of Korea (NRF) Grant funded by the Korean Government (MSIP) (No. 2016R1D1A3B03934797), the Ministry of Science and ICT (No. 2018R1A2A3075668), funded by the Ministry of Science, ICT & Future Planning (No. 2014R1A5A1009799), and funded by the Ministry of Science the Technology Innovation Program (10070150) funded by the Ministry of Trade, Industry & Energy (MI, Korea).

**Conflicts of Interest:** The authors declare no conflict of interest.

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
