# Peer review of "High-Throughput Production of Heterogeneous RuO2/Graphene Catalyst in a Hydrodynamic Reactor for Selective Alcohol Oxidation"

_catalysts, doi:10.3390/catal9010025_

Round 1

Reviewer 1 Report

The manuscript “catalysts-413542by Kim et al. entitled "High throughput production of heterogeneous Ru/graphene catalyst in a hydrodynamic reactor for selective alcohol oxidation" reports on the synthesis and characterization of a ruthenium-based catalyst, together with its use in the selective alcohol oxidation to aldehyde. The reported results could be of interest for the readers of Catalysts, after the following major revisions.

Pag 3, row 100: authors claim: “The mole ratio of Ru on Ru/IL-graphene is 2.5 mol%”. I think it should be much more useful for catalytic purposes to assess the Ru%w in the catalyst by elemental analysis. Otherwise, the authors should clarify how they calculated the molar ratio in the catalyst, being the support composed by both graphene and IL.

Pag. 4, figure 5: XPS spectra show oxidation state +4 for Ru (RuO2). For this reason, it is not correct talking about Ru nanoparticles throughout the text, because a reader could think about Ru(0) nanoparticles, which is not the case.

Pag. 6, tables 1 and 2: in the table caption it is reported Ru = 2.5 mol%. This molar ratio is referred to the substrate (0.5 mmol) or, again, to Ru/IL-graphene molar ratio? In the latter case, the author should report the value of Ru/benzyl alcohol molar ratio employed in the catalytic runs.

Pag. 8, row 219: “The isolated catalyst was washed with an aqueous solution of NaOH (1M) and then dried in vacuo before being recycled.” Why the authors washed their catalyst with NaOH solution? Generally, aerobic oxidation of alcohols needs basic conditions. Many authors add Na2CO3 to the reaction mixture, which is cheaper than NaOH. Some catalytic systems do not use any base at all. What happens if the authors wash the isolated catalyst with methanol, or simply with water (without any base)?

Pag. 5: Ru%w in the catalyst before and after 5 runs should be measured to check for metal leaching into solution.

A minor point: add some recent references on alcohol aerobic oxidation under green conditions, such as Catalysts 2018, 8, 431, and t. J. Mol. Catal. A Chem. 2014, 386, 114–119.

Author Response

Response to the Reviewer 1’s Comments

Reviewer #1: The manuscript “catalysts-413542” by Kim et al. entitled "High throughput production of heterogeneous Ru/graphene catalyst in a hydrodynamic reactor for selective alcohol oxidation" reports on the synthesis and characterization of a ruthenium-based catalyst, together with its use in the selective alcohol oxidation to aldehyde. The reported results could be of interest for the readers of Catalysts, after the following major revisions.

1. Comment: Pag 3, row 100: authors claim: “The mole ratio of Ru on Ru/IL-graphene is 2.5 mol%”. I think it should be much more useful for catalytic purposes to assess the Ru%w in the catalyst by elemental analysis. Otherwise, the authors should clarify how they calculated the molar ratio in the catalyst, being the support composed by both graphene and IL.

Our response: Thank you for this comment. As reviewer suggested, we recalculated the weight percentage of Ru in the catalyst by elemental analysis. And, we changed the related texts as below.

Changes in the manuscript

[On pages 3]

2.5 mol% 1.5 wt%

[On pages 8]

2.5 mol% 1.5 wt%

2. Comment: Pag. 4, figure 5: XPS spectra show oxidation state +4 for Ru (RuO2). For this reason, it is not correct talking about Ru nanoparticles throughout the text, because a reader could think about Ru(0) nanoparticles, which is not the case.

Our response: Thank you for this comment; it has helped us improve our manuscript. As reviewer suggested, we entirely modified the “Ru” to the “RuO2” in our original manuscript.

3. Comment: Pag. 6, tables 1 and 2: in the table caption it is reported Ru = 2.5 mol%. This molar ratio is referred to the substrate (0.5 mmol) or, again, to Ru/IL-graphene molar ratio? In the latter case, the author should report the value of Ru/benzyl alcohol molar ratio employed in the catalytic runs.

Our response: To identify loading level of ruthenium on RuO2/IL-graphene, it was extracted using by strong acid, then the amount of ruthenium was analyzed by inductive coupled plasma spectroscopy (ICP-AES). Through the ICP analysis, we indicated the 2.5 mol% of Ru in the catalyst. We are very regret confusing you. We corrected 2.5 mol% to 1.5 wt% in Table 1 and Table 2 by recalculating the weight percentage of Ru as you pointed out in page 3.

Changes in the manuscript

[On pages 6]

Table 1 caption: 2.5 mol% 1.5 wt%

[On pages 7]

Table 2 caption: 2.5 mol% 1.5 wt%

4. Comment: Pag. 8, row 219: “The isolated catalyst was washed with an aqueous solution of NaOH (1M) and then dried in vacuo before being recycled.” Why the authors washed their catalyst with NaOH solution? Generally, aerobic oxidation of alcohols needs basic conditions. Many authors add Na2CO3 to the reaction mixture, which is cheaper than NaOH. Some catalytic systems do not use any base at all. What happens if the authors wash the isolated catalyst with methanol, or simply with water (without any base)?

Our response: Thank you for this comment a lot. As reviewer suggested, the isolated catalyst was just washed with water without NaOH (1M) solution and then dried in vacuo before being recycled. Using the catalyst during recycling test, the selectivity and conversion was kept over several cycles. So, NaOH treatment is not required for washing step. In the manuscript, we corrected “with an aqueous solution of NaOH (1M)” to “with water simply”.

5. Comment: Pag. 5: Ru%w in the catalyst before and after 5 runs should be measured to check for metal leaching into solution.

Our response: As you suggested, we had checked Ru wt% in the catalyst by inductive coupled plasma spectroscopy after 5 number cycles. There are no metal leaching into filtration solution. Our catalyst showed excellent substrate conversion and yield of corresponding product.

6. Comment: A minor point: add some recent references on alcohol aerobic oxidation under green conditions, such as Catalysts 2018, 8, 431, and t. J. Mol. Catal. A Chem. 2014, 386, 114–119.

Our response: We changed Ref 10 and 11 to recent references which you suggested as following.

Chan-Thaw, C. E.; Savara, A.; Villa A. Selective Benzyl Alcohol Oxidation over Pd Catalysts, Catalysts 2018, 8, 431. DOI: 10.3390/catal8100431.

Dell’Anna, M. M.; Mali, M.; Mastrorilli, P.; Monopoli, A. Oxidation of benzyl alcohols to aldehydes and ketones under air in water using a polymer supported palladium catalyst, J. Mol. Catal. A Chem. 2014, 386, 114-119. DOI: 10.1016/j.molcata.2014.02.001.

Reviewer 2 Report

The manuscript entitled “High-throughput production of heterogeneous Ru/graphene catalyst in a hydrodynamic reactor for selective alcohol oxidation” has been examined and in my opinion it is an interesting work and carefully prepared, so it can be published in present form, with only a few minor changes that I indicate below:

1 - The authors should increase Figure 2 because some images are not clear;

2 - In table 1 authors should indicate the meaning of entry 5 as they did for entries 1-4;

3 - In both Table 1 and Table 2, authors should indicate how yield and selectivity were calculated.

Author Response

Response to the Reviewer 2’s Comments

Reviewer #2: The manuscript entitled “High-throughput production of heterogeneous Ru/graphene catalyst in a hydrodynamic reactor for selective alcohol oxidation” has been examined and in my opinion it is an interesting work and carefully prepared, so it can be published in present form, with only a few minor changes that I indicate below:

1. Comment: The authors should increase Figure 2 because some images are not clear;

Our response: Thank you for this helpful comment. As reviewer suggested, we increased Figure 2 in original manuscript.

2. Comment: In table 1 authors should indicate the meaning of entry 5 as they did for entries 1-4;

Our response: Thank you for this comment. As reviewer’s suggestion, we have modified the entry 5 as follows:

Changes in the manuscript

[On pages 6]

Table 1 caption

Ru(OH)x/SNP Ru(OH)x/SNP[e]

[e] Ru supported on silica nano particles

3. Comment: In both Table 1 and Table 2, authors should indicate how yield and selectivity were calculated.

Our response: Thank you for this helpful comment. According to reviewer’s suggestion, Table 2 already contains the calculation method and we modified the Table 1 as follows:

[On pages 6]

Table 1 caption

Conversion and selectivity were calculated by GC and GC-MS analysis by using biphenyl as an internal standard.

Round 2

Reviewer 1 Report

The manuscript could be accepted for publication, after the following minor corrections.

Row 101, pag 3: please convert "mole ratio" into "percent amount";

In paragraph 3.4 (Catalytic Reaction) and in table 2 caption, please, specify the amount (in mg) of RuO2/IL-graphene used.

Author Response

Response to the Reviewer 1’s Comments-2nd

Reviewer #1: The manuscript could be accepted for publication, after the following minor corrections.

In paragraph 3.4 (Catalytic Reaction) and in table 2 caption, please, specify the amount (in mg) of RuO2/IL-graphene used.

1. Comment: Row 101, pag 3: please convert "mole ratio" into "percent amount"

Our response: Thank you for helpful comment. We corrected "mole ratio" to "percent amount" in row 101, page 3.

Changes in the manuscript

[On pages 3]

molar ratio percent amount

2. Comment: In paragraph 3.4 (Catalytic Reaction) and in table 2 caption, please, specify the amount (in mg) of RuO2/IL-graphene used.

Our response: As you point out, we added the amount (unit: mg) of RuO2/Il-graphene in Table 2 and in paragraph 3.4 Catalytic reaction

Changes in the manuscript

[On pages 7]

RuO2/Il-graphene (Ru = 1.5 wt%) RuO2/Il-graphene (Ru = 1.5 wt%, 10 mg)

[On pages 8]

RuO2/Il-graphene catalyst (Ru: 1.5 wt%) RuO2/Il-graphene catalyst (Ru: 1.5 wt%, 10 mg)